

# A low caffeine dose improves maximal strength, but not relative muscular endurance in either heavier-or lighter-loads, or perceptions of effort or discomfort at task failure in females

Georgina Waller, Melissa Dolby, James Steele and James P. Fisher

School of Sport Health and Social Sciences, Solent University, Southampton, United Kingdom

## ABSTRACT

**Background:** The body of literature considering caffeine as an ergogenic aid has primarily considered typically aerobic based exercise, male participants and moderate-to large-caffeine doses. With this in mind the aim of this project was to explore the effects of a low-caffeine dose upon maximal voluntary contraction (MVC) and muscular endurance (time to task failure, TTF) at heavier-and lighter-loads.

**Methods:** Nineteen physically active, habitual caffeine consuming females randomly performed four testing conditions; two with a low-dose of caffeine (100 mg equating to mean = $1.5 \pm 0.18$ mg·kg$^{-1}$) and two placebo conditions, where they performed a maximal strength test (MVC) knee extension at 45° followed by a task of relative muscular endurance (sustained isometric contraction for TTF) using either heavier-(70% MVC) and lighter-(30% MVC) loads. Each participant performed each load condition following both caffeine and placebo consumption. Immediately following cessation of the muscular endurance test participants were asked to report their rating of perceived effort (RPE) and rating of perceived discomfort (RPD).

**Results:** Analyses revealed a significant effect for caffeine upon MVC compared to placebo ($p = 0.007$). We also found a significantly greater TTF for the lighter-compared to the heavier-load condition ($p < 0.0001$); however, there was no significant effect comparing caffeine to placebo ($p = 0.2368$), but insufficient precision of estimates to infer equivalence in either lighter-($p = 0.750$) or heavier-load ($p = 0.262$) conditions. There were no statistically significant effects for caffeine compared with placebo, or lighter-compared with heavier-loads, for RPE and RPD (all $p > 0.05$). RPE was statistically equivalent between caffeine and placebo for both lighter-($p = 0.007$) and heavier-load ($p = 0.002$) conditions and RPD for heavier-($p = 0.006$) but not lighter-load ($p = 0.136$).

**Discussion:** This is the first study to demonstrate a positive effect on strength from a low caffeine dose in female participants. However, it is unclear whether caffeine positively impacts upon relative muscular endurance in either heavier-or lighter-loads. Further, both RPE and RPD appear to be relatively similar during isometric tasks performed to task failure independently of caffeine supplementation or load.

Corresponding author
James P. Fisher,
james.fisher@solent.ac.uk

These findings may have implications for persons wishing to avoid side-effects or withdrawal symptoms associated with larger caffeine doses whilst still attaining the positive strength responses.

## INTRODUCTION

Caffeine is evidenced to increase productivity, improve concentration and reduce fatigue (*Bazzucchi et al., 2011*; *Pickering & Kiely, 2018*) and as such, in an effort to enhance performance, it is reported that ~74% of athletes consume caffeine as an ergogenic aid prior to competition (*Del Coso, Muñoz & Muñoz-Guerra, 2011*). Historically the body of research has considered the effects of caffeine upon predominantly aerobic endurance type exercise, for example time to exhaustion in well-trained and elite cyclists (*Costill, Dalsky & Fink, 1978*; *Pasman et al., 1995*), 2,000 m rowing performance (*Bruce et al., 2000*) and 1,500 m treadmill running (*Wiles et al., 1992*). Indeed, a recent umbrella review of published meta-analyses considering the effects of caffeine supplementation on a range of performance outcomes found that evidence was stronger for predominantly aerobic vs. anaerobic tasks (*Grgic et al., 2019a*). Yet, perhaps in part due to the popularity of caffeinated supplements aimed to harness its aforementioned benefits, caffeine is commonly consumed as part of a 'pre-workout' with the desire to enhance strength, power and muscular endurance before partaking in resistance exercise or strength/power-based tasks (*Cesareo et al., 2019*). In comparison with aerobic endurance tasks, there is still a gap within the present body of literature considering caffeine consumption and muscular strength/endurance tasks. Further, as with most resistance training studies, the body of research considering the maximal strength and muscular endurance effects of caffeine have dominantly considered trained or untrained male participants.

Indeed, a recommendation from recent reviews by *Grgic et al. (2019a, 2019b)* was for more research to be conducted in women. This is supported by evidence suggesting sex-differences in subjective and physiological responses to caffeine consumption (*Temple & Ziegler, 2011*). To date research considering female participants has used large caffeine doses. For example *Goldstein et al. (2010)* considered 15 females, reporting a small yet significantly ($p < 0.05$) greater barbell bench press 1RM for caffeine (6 mg·kg$^{-1}$) compared to placebo conditions (52.9 ± 11.1 kg vs. 52.1 ± 11.7 kg, respectively); and *Sabblah, Dixon & Bottoms (2015)*, who compared caffeine (5 mg·kg$^{-1}$) and placebo for strength differences between males ($n = 10$) and females ($n = 8$). The authors reported significantly greater 1RM bench press following caffeine compared to placebo, ($p = 0.016$; males = 101.5 ± 28.9 to 107.5 ± 30.5 kg, females = 32.2 ± 9.0 to 35.3 ± 7.3 kg) with no between sex differences. In addition, they reported a tendency for an increase in relative muscular endurance (repetitions at 40% 1RM; $p = 0.059$) with a between sex difference of $p = 0.061$ in favour of males compared to females as a result of caffeine

ingestion. Finally, *Sabblah, Dixon & Bottoms (2015)* found no improvement in squat 1RM for either males or females for the caffeine condition. The lack of research considering female participants might simply be a reflection that the bulk of strength training research considers male participants. However, it might also be a product of the sensitivity around female menstrual cycles and oral contraceptive use which might impact the effects of caffeine upon the body (*Abernethy & Todd, 1985*).

A further limitation of present literature is a lack of investigation into low caffeine dosage ($<3$ mg$\cdot$kg$^{-1}$) for strength measures (*Spriet, 2014*). Indeed, *Grgic et al. (2019a)* note that the doses of caffeine across the body of literature have generally been around 6 mg of caffeine per kilogram of body mass (mg$\cdot$kg$^{-1}$) or greater. Some studies suggest that lower doses of caffeine could be impactful upon strength performance. For example, *Arazi, Hoseinihaji & Eghbali (2016)* reported maximal strength (leg press 1RM) across two caffeine conditions (2 mg$\cdot$kg$^{-1}$ and 5 mg$\cdot$kg$^{-1}$) as well as a placebo condition. Whilst the analyses didn't reveal a statistically significant difference, their descriptive data (leg press 1RM; placebo = 154 $\pm$ 27.3 kg vs. 2 mg$\cdot$kg$^{-1}$ of caffeine = 171 $\pm$ 15.9 kg) seems noteworthy. A more recent study of the dose-response effects of 2 mg$\cdot$kg$^{-1}$, 4 mg$\cdot$kg$^{-1}$ and 6 mg$\cdot$kg$^{-1}$ of caffeine upon both upper and lower body strength and relative muscular endurance in males reported a linear trend for upper body strength, but no clear dose-response relationship for lower body strength or relative muscular endurance (*Grgic et al., 2020*). Thus, there is the possibility that low doses of caffeine may still produce performance enhancing effects. This might be particularly important considering studies reporting negative side effects to caffeine supplementation. For instance, *Goldstein et al. (2010)* reported that 20% of participants 'exhibited intense emotional side effects, including an expressed inability to verbally communicate, focus, and/or remain still as well as the feeling of wanting to cry' (page 4). Other studies have negative side-effects following caffeine withdrawal, including severe fatigue, increased muscle pain and cramps, sleep disturbance, irritability, headaches, and occasional nausea (*Nehlig, 1999*; *Juliano & Griffiths, 2004*). In review, withdrawal symptoms were evident from a median dose of 357 mg (in a 60 kg female and 80 kg male this equates to 5.95 mg$\cdot$kg$^{-1}$ and 4.46 mg$\cdot$kg$^{-1}$, respectively) but also in doses as low as 129 mg (in the same example: 2.15 mg$\cdot$kg$^{-1}$ and 1.61 mg$\cdot$kg$^{-1}$, respectively; *Strain et al., 1994*).

Finally, an additional gap within the body of research is that to date no empirical publications have considered the impacts of caffeine supplementation upon the effects of heavier-compared with lighter-load muscular endurance tasks. These might differ following caffeine consumption as a result of the effects of caffeine on the neurological system and the potentiation of force production during submaximal intensities (*Tarnopolsky, 2008*), as well as the potentially differing effects of caffeine on slow twitch muscle fibres (*Davis & Green, 2009*). Current research suggests that lighter-load resistance exercise tasks, whilst producing similar perceptions of effort at momentary task failure, produce greater discomfort compared to heavier loads (e.g. *Fisher & Steele, 2017*; *Stuart et al., 2018*). However, these studies did not consider the use of caffeine ingestion which might improve endurance performance through a reduction in perceived effort at submaximal exercise intensities thus deterring task disengagement due to maximal

perceived effort for longer (*Duncan et al., 2013*; *De Morree, Klein & Marcora, 2014*; *Smirnaul et al., 2017*). It has also been shown that caffeine produces analgesic effects and thus may improve performance similarly by reducing pain perception allowing higher exercise intensities to be produced at the same level of pain perception, or again deterring task disengagement due to reaching maximum tolerable pain levels (*Motl, O'Connor & Dishman, 2003*; *Motl et al., 2006*; *Astorino et al., 2011*; *Duncan et al., 2013*). Indeed, relative muscular endurance typically does not change with interventions including resistance training, but where it does it has been speculated this may be due to reductions in perceived discomfort (*Fisher et al., 2020*). However, both the perceived effort and perceived pain reducing effects of caffeine have only been shown with higher doses (4–10 mg·kg$^{-1}$).

With the above in mind, the aim of this study was to consider the effects of a low caffeine dose (100 mg/<2 mg·kg$^{-1}$) compared to a placebo upon maximal strength and relative muscular endurance tasks, in addition to perceptions of effort and discomfort, at both heavier-and lighter-loads (70% and 30% MVC, respectively) in a female population. In this study design we have utilised isometric knee extension strength testing, and time to task failure (TTF) at %MVC. We hypothesised that caffeine would positively impact MVC, as well as TTF at 30% MVC and be related to the perceptual effects of caffeine.

## METHODS

### Experimental approach to the problem

The present study used a double-blind, randomised control and crossover counterbalanced research design whereby all participants completed caffeine and placebo conditions for maximal strength testing (MVC) of the knee extensors and both heavier-(70% MVC) and lighter-(30% MVC) load isometric knee extension TTF (e.g. all participants performed four conditions).

### Participants

Following approval from Solent University Health, Exercise and Sport Science ethics committee (ethical application reference number: wallg2018), 19 physically active female participants (age range 18–35 years, mean (±SD) age = 21.6 ± 3.8 years, height = 1.68 ± 0.07 cm, mass = 67.9 ± 8.4 kg, body mass index = 24.2 ± 2.8) were recruited from a University campus. All participants were considered healthy based upon responses to a modified physical activity readiness questionnaire. Inclusion criteria required that all participants be habitual caffeine drinkers (>100 mg/day and recorded via a 4-day food diary based on *Motl et al., 2006*) and be free from musculoskeletal injuries for which a knee extension task might be contraindicated, as well as not currently undertaking a structured resistance training programme. All participants were provided a verbal briefing about the study, provided a participant information sheet for their records, and completed an informed consent form. Exclusion criteria included being pregnant, breast feeding (<6 months), consuming androgenic anabolic steroids, diabetic, hypertensive, consuming >300 mg/day of caffeine.

## Familiarisation

Prior to testing, each participant attended the laboratory where height and mass were recorded, and paperwork were completed. The isokinetic dynamometer (Humac Norm, CSMi, Stoughton, MA, USA) was set up to ensure participant comfort, whilst the lateral epicondyle of the right knee was aligned with the axis of rotation on the device. The cushioning pad of the leg extension was positioned central to the shin and seat and position measurements were recorded for replication for each condition for each participant. All participants confirmed right leg dominance when asked with which leg they would kick a football with and thus, the right leg was used for testing for all conditions. Following set-up each participant completed MVC testing at 45° of knee extension and the lighter load (30% MVC) condition as outlined below (data was not recorded) to provide familiarisation to isokinetic testing and isometric TTF. During the familiarisation session participants were also informed of the use of rating of perceived exertion-effort and rating of perceived exertion and perceived discomfort scales (*Steele et al., 2017*). The scales and instructions are available on the Open Science Framework (https://osf.io/ufvy8/).

## Blinding, randomisation and dosage

A person external to the project was asked to label both the caffeine and placebo tablet as supplement 'A' or 'B' and deposit each into an opaque container to ensure double blinding from both the researchers providing the supplement and the participants. The caffeine consisted of $2 \times$ Pro-Plus Tablets (a total of 100 mg caffeine), and the placebo was $2 \times$ Deva Vegan D2 Vitamin D 1200IU tablet (0.06 mg D2). Vitamin D2 was selected based on similarity and size to the caffeine tablet and since studies have shown this to be ineffective upon muscular strength (*Chiang et al., 2017*). For the caffeine conditions the dosage of caffeine was maintained at 100 mg per person since this represents ecological validity for consumption (e.g. we believe a person is likely to consume the dosage suggested on the packet not calculate a specific dosage based on their body mass). The dosage provided resulted in a body mass relative dose at $1.50 \pm 0.18$ mg·kg$^{-1}$ (range = 1.1 to 1.85 mg·kg$^{-1}$).[1]

[1] For context and comparison, an 8.4fl.oz/250 ml can of Red Bull contains 80 mg of caffeine, equal to 1.17 mg·kg$^{-1}$ for the present participants.

## Testing

On the morning of testing, participants had been instructed not to consume caffeine-containing food or beverages. A sheet including drinks, food and medication to avoid was provided in advance to participants (in brief this included; coffee, tea, chocolate, carbonated beverages, energy drinks of any kind, caffeine/energy gels, chewing gum, supplement bars including protein and energy bars, weight-loss tablets, pain-relieving tablets including codeine, paracetamol, ibuprofen, etc. and alcohol). Participants confirmed they had not participated in strenuous exercise for 48-h prior to testing for each condition. Participants were not requested to attend the testing sessions in a fasted state but were asked to replicate any foods consumed prior to the first testing condition for the latter conditions. Participants attended the laboratory for testing between 8 am and 11 am, and testing time was standardized for each participant across all conditions.

Participants returned no less than 6 days following the familiarisation for the first testing session to eliminate the effects of any delayed onset muscle soreness. Each participant received either supplement 'A' or 'B' (selected at random via a random letter generator) and was asked to consume it together with 150 ml of tap water (*Smit & Rogers, 2000*). After 60 min, during which no food consumption or physical activity was permitted, and toilet breaks and water consumption were logged, the participant was asked to complete a brief 5-min warm-up on a cycle ergometer (Monark, Ergomedic 874e) up to 60% age-predicted maximum heart rate to warm key muscles (*Fisher & Steele, 2017*). Following this, the participant completed a maximal voluntary contraction (MVC) at 45° knee extension. Participants were instructed to gradually build up to a maximal effort over 3 s and were instructed to gradually reduce their effort once it was clear that a max torque had been achieved (i.e. when the torque reading was no longer increasing). After 30 s rest, the participant completed the isometric leg extension task to failure. The lever arm was maintained at 45° knee extension and the participant was provided visual feedback on a display screen of what force to maintain (70% or 30% for heavier-and lighter-load conditions, respectively) with an upper and lower bar of ±5% (e.g. if MVC was 100 Nm then for the 70% condition a bar was shown at 70 Nm with an upper and lower bar 3.5 Nm above and below, respectively). Verbal encouragement was provided throughout to ensure maximal exerted effort (*Timmins & Saunders, 2014*), and if they fell below the bottom torque limit then they were encouraged to attempt to regain their set torque output. The task was ceased when the participant could not maintain the required force despite attempting to with maximal effort. Rating of perceived effort (RPE) and rating of perceived discomfort (RPD) was measured and recorded immediately following the isometric knee extension task (*Steele et al., 2017*) along with TTF (seconds).

For the successive three testing sessions with no less than 6 days recovery between, participants repeated the testing process as described above. Each participant was randomly assigned a supplement and condition from their remaining allocation, for example each participant had 2 × 'A' and 2 × 'B' supplements, for both heavier-and lighter-load conditions (70% and 30% MVC, respectively).

## Statistical analysis

The independent variables were the training load (70% and 30% MVC) and the supplement condition (placebo and caffeine). The dependent variables included MVC, TTF, RPE and RPD. All analysis was done using the 'lme4' package (*Bates et al., 2015*) and 'lmerTest' in R (v3.6.1; R Foundation for Statistical Computing, Vienna, Austria. URL http://www.R-project.org/). Linear mixed modelling was used to analyse all outcomes using Restricted Maximum Likelihood estimation with participants as a level 3 variable, 'Load' (70% and 30% MVC) as a level 2 variable, and 'Condition' (placebo and caffeine) as a level 1 variable. 'Load' and 'Condition' were modelled as fixed factors with random intercepts by participants included. For MVC only Condition as examined as a factor with four separate observations per participant and two per condition due to this outcome being measured at the beginning of every trial. Estimated marginal means with 95% confidence intervals were produced using the 'emmeans' package from each model
(for each Condition for MVC, and for each Condition by Load for all other dependent variables) in addition to pairwise contrasts. Contrasts were also performed with an equivalence testing approach and 90% confidence intervals. Equivalence bands were determined from reliability data for MVC taken from our laboratory using the same set-up as in the present study. The half-width of the minimal detectable change % (6.98%) was used. Because we did not have similar reliability data for TTF we opted to use the same relative minimal detectable change value for this dependent variable also in setting equivalence bands. The absolute equivalence bands for MVC and TTF were ±6.45 Nm and ±14.25 s respectively. For RPE and RPD we used the half-width of the minimal detectable change determined from previously published reliability data for these scales (*Steele et al., 2017*). The absolute equivalence bands for RPE and RPS were ±1.18 and ±1.40 respectively. All tests were conducted with α = 0.05 for determination of statistical significance. In addition, data visualisation included plotting individual raw data and estimated marginal means for repeated measures between condition, in addition to pairwise contrasts with both 95% and 90% confidence intervals and equivalence bands. Standardised effect sizes were calculated as Cohen's *d* (*Cohen, 1992*), and interpreted using Cohen's thresholds (>0.2 to <0.5 'small'; >0.5 to <0.8 'moderate'; >0.8 'large') using the eff_size() function in the emmeans package.

## RESULTS

### Maximal voluntary contraction

A statistically significant main effect of Condition was found in the linear mixed model for MVC ($F_{(1, 56)} = 7.906$, $p = 0.007$) with a higher MVC for caffeine (caffeine = 92.8 Nm (78.5 to 107.0 Nm), placebo = 82.6 Nm (68.4 to 96.8 Nm)). Cohen's *d* for the between condition contrast was 0.65 (0.15 to 1.14). Equivalence testing did not reveal a statistically significant effect ($t_{(56)} = -1.141$, $p = 0.129$) for the pairwise contrast. Visual inspection of Fig. 1 reveals that traditional hypothesis testing excludes a null hypothesis of 0 Nm difference between caffeine and placebo with a greater MVC seen for caffeine. However, the 95% confidence intervals do not exclude the upper limit of the equivalence bands and so it is unclear whether there is a meaningful effect of caffeine upon MVC.

### Time to task failure

There was no statistically significant main effect of Condition found in the linear mixed model for TTF ($F_{(1, 54)} = 1.431$, $p = 0.2368$) though there was a significant main effect of Load ($F_{(1, 54)} = 1.431$, $p < 0.001$) with greater TTF for 30% MVC (caffeine(30% MVC) = 204.1 s (172.1 to 236.1 s), placebo(30% MVC) = 178.2 s (146.2 to 210.2 s), caffeine (70% MVC) = 65.3 s (33.3 to 97.3 s), placebo(70% MVC) = 62.1 s (30.0 to 94.1 s)). There was also no statistically significant interaction effect of Condition × Load ($F_{(1, 54)} = 0.870$, $p = 0.355$). Cohen's *d* for the between condition contrast was 0.49 (−0.17 to 1.15) in 30% MVC and 0.06 (−0.59 to 0.71) in 70% MVC. Equivalence testing also did not reveal a statistically significant effect for Condition either in 30% MVC ($t_{(54)} = 0.679$, $p = 0.750$) or 70% MVC ($t_{(54)} = -0.640$, $p = 0.262$) for the pairwise contrasts. Visual inspection of Fig. 2 reveals that traditional hypothesis testing fails to exclude a null

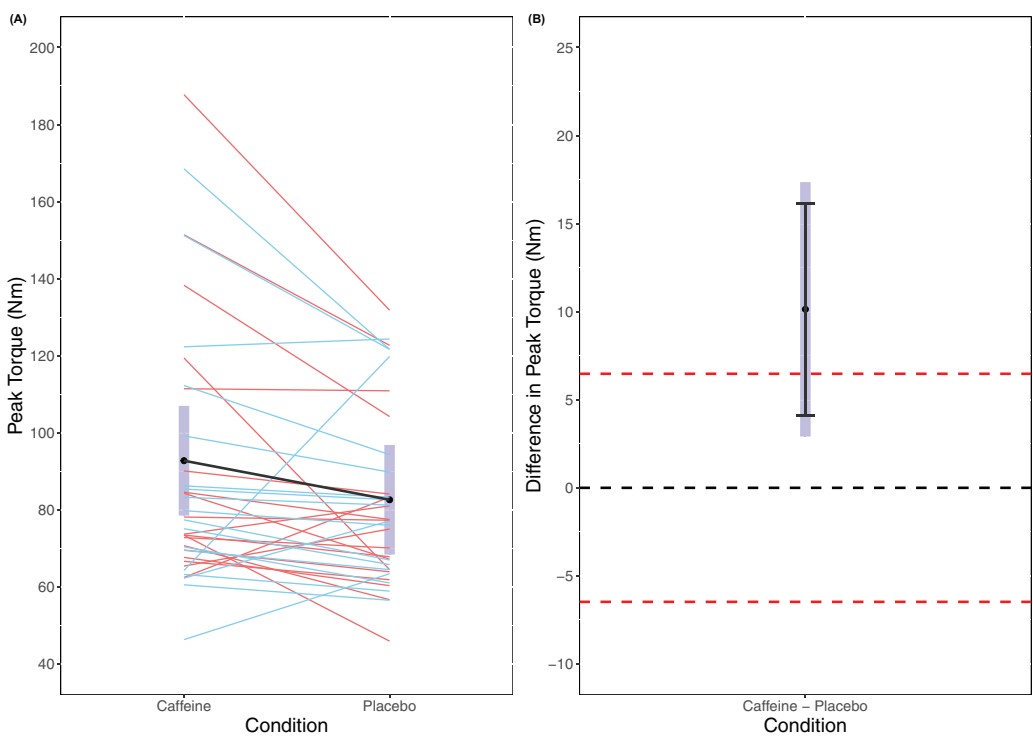

**Figure 1 Peak Torque (A) and difference in peak torque (B) between caffeine and placebo conditions.**

hypothesis of 0 Nm difference between caffeine and placebo, yet the interval estimates are insufficiently precise to infer equivalence between the conditions.

## Rating of perceived effort

There was no statistically significant main effect of Condition found in the linear mixed model for RPE ($F_{(1, 54)} = 0.518$, $p = 0.475$) or of Load ($F_{(1, 54)} = 3.056$, $p = 0.086$), and no statistically significant interaction effort of Condition × Load ($F_{(1, 54)} = 0.095$, $p = 0.759$) with RPE appearing similar across all trials (caffeine(30% MVC) = 8.26 (7.65 to 8.87), placebo (30% MVC) = 8.00 (7.39 to 8.61), caffeine(70% MVC) = 8.63 (8.02 to 8.24), placebo(70% MVC) = 8.53 (7.92 to 9.14)). Cohen's *d* for the between condition contrast was 0.24 (−0.42 to 0.89) in 30% MVC and 0.09 (−0.56 to 0.74) in 70% MVC. Equivalence testing revealed a statistically significant effect for Condition in both 30% MVC ($t_{(54)} = -2.528$, $p = 0.007$) and 70% MVC ($t_{(54)} = -2.964$, $p = 0.002$) for the pairwise contrasts. Visual inspection of Fig. 3 reveals that across both 30% MVC and 70% MVC there was equivalence in RPE between caffeine and placebo.

## Rating of perceived discomfort

There was no statistically significant main effect of Condition found in the linear mixed model for RPE ($F_{(1, 54)} = 1.772$, $p = 0.189$) or of Load ($F_{(1, 54)} = 0.088$, $p = 0.769$), and no statistically significant interaction effort of Condition × Load ($F_{(1, 54)} = 1.072$, $p = 0.305$) with RPD appearing similar across all trials (caffeine(30% MVC) = 8.00 (7.26 to 8.74),

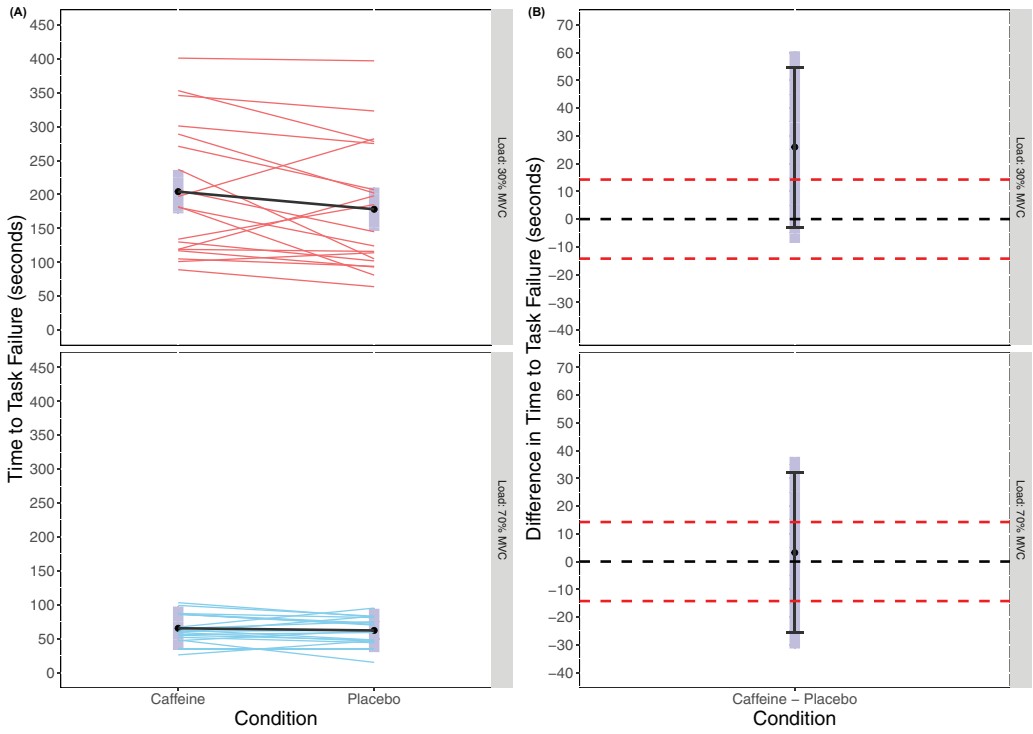

**Figure 2 Time to task failure (A) and difference in time to task failure (B) for lighter (30% MVC) and heavier (70% MVC) load and between caffeine and placebo conditions.**

placebo (30% MVC) = 7.16 (6.42 to 7.89), caffeine (70% MVC) = 7.53 (6.79 to 8.26), placebo (70% MVC) = 7.42 (6.69 to 8.16)). Cohen's $d$ for the between condition contrast was 0.54 (−0.11 to 1.20) in 30% MVC, and 0.07 (−0.58 to 0.72) in 70% MVC. Equivalence testing revealed a statistically significant effect for Condition in the 70% MVC ($t_{(54)}$ = −2.572, $p$ = 0.006) but not 30% MVC ($t_{(54)}$ = −1.108, $p$ = 0.136) for the pairwise contrasts. Visual inspection of Fig. 4 reveals that for 70% MVC there was equivalence in RPD between caffeine and placebo.

## DISCUSSION

The present study considered the effects of low doses of caffeine upon maximal force (MVC) and TTF at 30% and 70% MVC in females. The aim of the study was to better understand whether low caffeine doses can impact maximal strength (MVC) performance as well as affect relative muscular endurance (TTF) and whether this was associated with an effect of reducing perceptions of effort or discomfort. To our knowledge this is the first study to consider the impact of low doses of caffeine (<2 mg·kg$^{-1}$) upon MVC and TTF for heavier-and lighter-loads in females. We believe our study represents an ecologically valid approach since people typically consume a predetermined absolute dosage of caffeine (e.g. 2 × Pro-Plus tablets; equating to a total of 100 mg caffeine) and rarely calculate a specific dosage based on their body mass. Our main findings were that 100mg caffeine (~1.5 mg·kg$^{-1}$) supported an increase in strength, but not relative
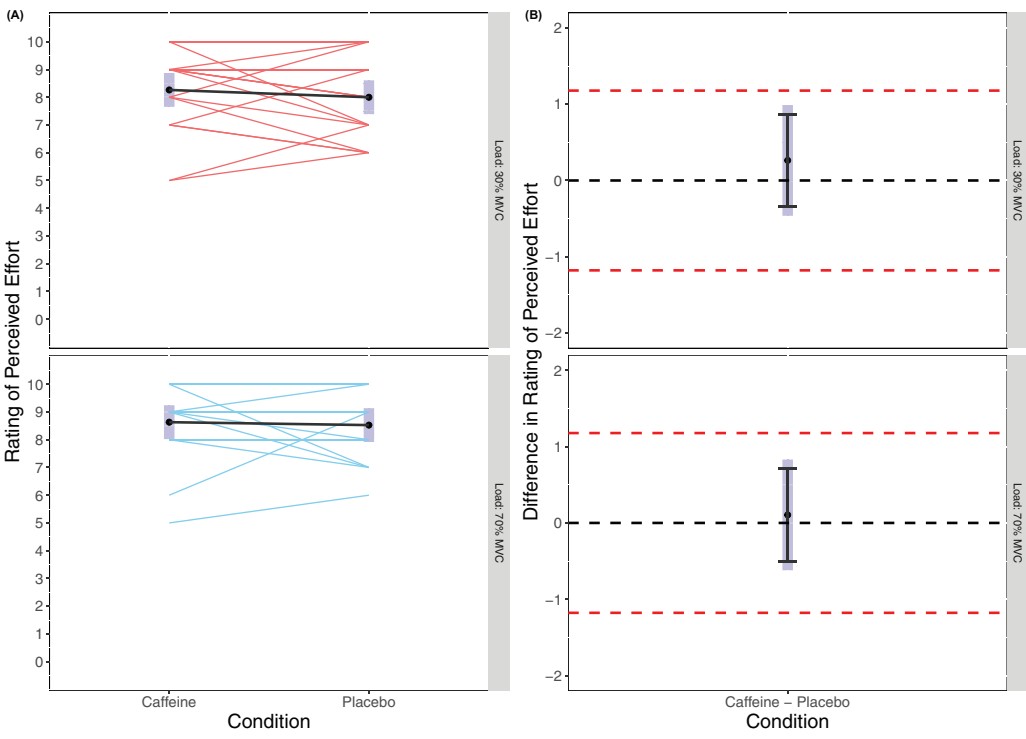

**Figure 3 Rating of perceived effort (A) and difference in rating of perceived effort (B) for lighter (30% MVC) and heavier (70% MVC) load and between caffeine and placebo conditions.**

muscular endurance compared to a placebo condition. These results are discussed in greater detail below.

## Maximal voluntary contraction

Our analyses revealed a significant effect for caffeine upon MVC in line with previous research showing positive effects upon strength as a result of caffeine ingestion (*Grgic et al., 2020*). Thus, even at low doses caffeine may have the potential to positively impact maximal force production. However, whilst analyses did reveal a statistically significant effect (i.e. excluding a zero effect), the confidence intervals did not exclude the upper equivalence bands (Fig. 1B) and therefore it is unclear how meaningful this increase in MVC is. However, the standardised effect size (Cohen's *d*) for the caffeine compared to placebo condition for MVC was moderate (*d* = 0.645 (0.155 to 1.14)) which interestingly is larger than that reported in meta-analyses considered in a recent umbrella review (*Grgic et al., 2019a*). Though, the inclusion of the upper equivalence band and the point and interval estimates of prior meta-analyses in the confidence interval range for our estimate suggests that the true population effect may be lower and questions its meaningfulness. Further, prior research examining maximal strength in females has shown generally trivial to small effects at best (*Goldstein et al., 2010*; *Sabblah, Dixon & Bottoms, 2015*). However, we should recognize that caffeine consumption in females might stimulate very individual responses to MVC based on both genetic factors and the use of an

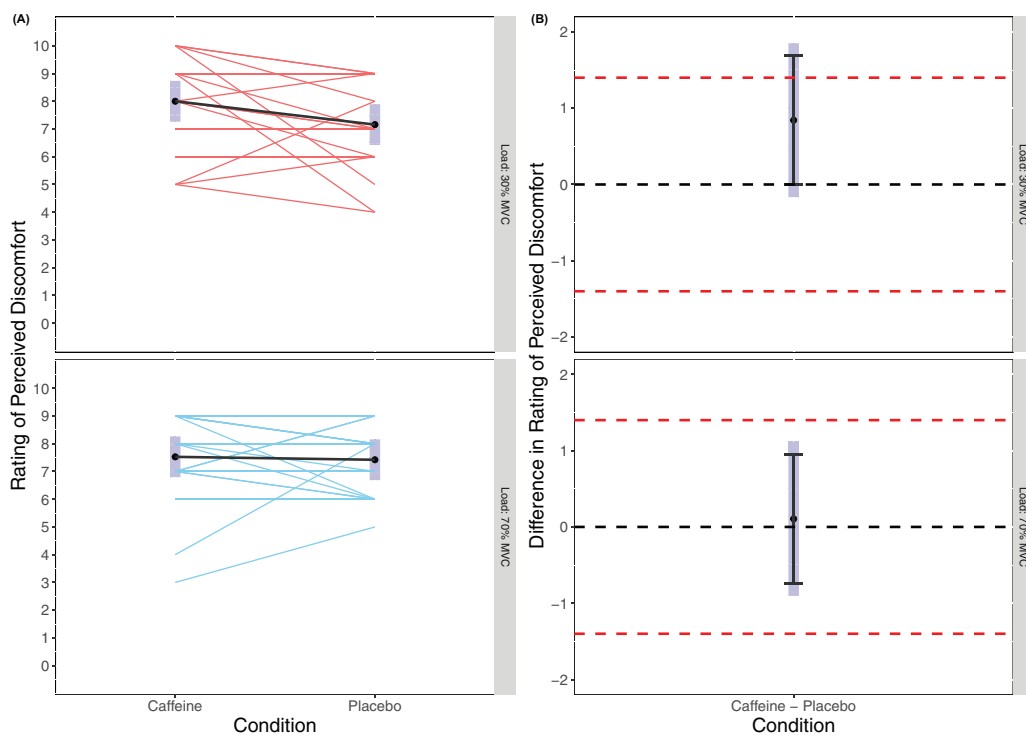

**Figure 4 Rating of perceived discomfort (A) and difference in rating of perceived discomfort (B) for lighter (30% MVC) and heavier (70% MVC) load and between caffeine and placebo conditions.**

oral contraceptive pill (*Guest et al., 2018*; *Abernethy & Todd, 1985*). Inspection of the individual participant data in Fig. 1 suggests there was considerable inter-individual variation. However, the extent to which this is true inter-individual variation in response to caffeine as opposed to merely test-retest variation is unclear and future research should consider employing replicated randomised cross-over designs to investigate this (*Atkinson, Williamson & Batterham, 2019*).

## Time to task failure

As expected, and supported by previous research, there was a statistically significantly longer TTF for the 30% MVC conditions compared to the 70% MVC conditions (30% MVC: caffeine = 204.1 s, placebo = 178.2 s; 70% MVC: caffeine = 65.3 s, placebo = 62.1 s). The body of literature has reported that when exercising with lighter loads a person can/must exercise for considerably longer to reach momentary failure (*Fisher & Steele, 2017*; *Stuart et al., 2018*). In contrast to previous research, our analyses revealed no statistically significant difference for TTF as result of caffeine compared to placebo ingestion. *Grgic et al. (2020)* recently reported significant increases in relative muscular endurance at 60% 1RM at all doses examined, though this study was in males only. *Sabblah, Dixon & Bottoms (2015)* found that, though not significant, any effect on relative muscular endurance at 40% 1RM appeared to be larger in males compared with females. It is worth considering however that with increases in strength resultant from

interventions such as resistance training, relative muscular endurance typically does not change (*Fisher et al., 2020*). In previous studies, strength is normally measured using 1RM on a separate day to the placebo and caffeine conditions and so where there are changes this may be due to increased strength from supplementation resulting in the load used for muscular endurance testing being relatively less. In the present study we tested MVC on the day of each condition and after supplementation. Thus, the lack of change in relative muscular endurance may be due to the fact that, though strength (MVC) increased, the load used for muscular endurance testing was set relative to that MVC.

However, while there was no statistically significant effect for Condition, or Condition × Load interaction, descriptively the lighter load (30% MVC) condition appears to differ between caffeine and placebo conditions (204.1 ± 97.32 s and 160.95 ± 81.31 s, respectively). While this was not statistically significant (neither excluding the upper equivalence bands nor a zero effect), a mean time change from placebo to caffeine of ~45 s might be practically meaningful. The standardised effect size for TTF following caffeine ingestion compared to placebo for the lighter load condition was moderate ($d = 0.489$ (−0.169 to 1.146)). This effect size is not too dissimilar from those reported in other meta-analyses though again the interval estimate suggests it may be lower (*Grgic et al., 2019a*). *Warren et al. (2010)* suggested that endurance improvements are the result of greater muscle fibre recruitment in the caffeine condition. However, the moderate effect for caffeine in the low load condition (30% MVC) was not evident in the higher load condition (70% MVC). At heavier loads electromyographic amplitude (a proxy for motor unit recruitment) is higher than at lighter loads (*Fisher, Steele & Smith, 2017*). As such, caffeine might be less likely to positively influence muscular endurance by motor unit recruitment in a heavier load condition. In contrast, at lighter loads caffeine ingestion might positively influence motor unit cycling and re-recruitment to sustain the force requirements, and thus enhance muscular endurance.

In addition, prior research has suggested that caffeine might impact on TTF due to its effects upon perceived effort and/or perceived pain (*Motl, O'Connor & Dishman, 2003*; *Motl et al., 2006*; *Astorino et al., 2011*; *Duncan et al., 2013*; *De Morree, Klein & Marcora, 2014*; *Smirnaul et al., 2017*). Previous studies comparing lighter-and heavier-load exercise to momentary failure have reported similar values for effort, but higher values for discomfort for a lighter load condition (*Fisher & Steele, 2017*; *Stuart et al., 2018*). Indeed, improvements in measures of relative muscular endurance have been speculated to be in part due to greater tolerance of perceived discomfort (*Fisher et al., 2020*). Thus, any performance enhancement in lighter-compared to heavier-load exercise may be due to effects upon perceptual variables. As such we measured both RPE and RPD.

## RPE and RPD/perceptual responses

Rating of perceived effort and Rating of perceived discomfort were recorded immediately after completing the knee extension task. There were no significant differences between caffeine and placebo conditions for RPE which, whilst not maximal in value (e.g. 10) for all participants, supports that the effort level was likely similar across all conditions. Indeed, equivalence testing was statistically significant suggesting that the RPEs reported

were equivalent across conditions and loads. This is somewhat contradictory of previous research which has reported lower RPE values following caffeine ingestion *during* exercise, though concordant with no differences in RPE at or after task failure (*Doherty & Smith, 2005*). This might be expected given that task failure should theoretically represent a ceiling for RPE and result in maximal or near maximal values irrespective of performance (even when affected by supplementation). Indeed, our results supported that RPE was statistically equivalent at task failure between conditions. *Doherty & Smith (2005)* however reported that the reduction in RPE during constant load exercise explained 29% of the variance in performance. Though a limitation of our study was that we did not measure RPE during each condition (and thus does not rule out RPE as potentially impacting TTF), a strength was the consideration of both ratings of perceived effort and discomfort where previous studies have generally failed to differentiate between the two. Since caffeine is shown to produce analgesic effects that may improve performance by reducing pain perception (*Motl, O'Connor & Dishman, 2003*; *Motl et al., 2006*; *Astorino et al., 2011*; *Duncan et al., 2013*), our data might simply support that effort was equivalent and unaffected by conditions independently of pain/discomfort perception.

Analyses of RPD also revealed no significant differences between heavier-or lighter-load conditions or between placebo or caffeine conditions. This could be interpreted to suggest that caffeine had no analgesic effects upon the perception of discomfort during prolonged exercise to momentary failure with a lighter-compared to heavier-load. That there were no differences between heavier-and lighter-load conditions (irrespective of caffeine or placebo conditions) suggests that an isometric task performed for time to failure might differ in the perceived discomfort incurred in comparison to performing traditional repetitions (e.g. both concentric and eccentric muscle actions). Certainly, previous research has supported differences in perceived discomfort between heavier-and lighter-load exercise (*Fisher & Steele, 2017*; *Stuart et al., 2018*). *Fisher & Steele (2017)* reported RPD values for dynamic knee extension exercise of 8.7 and 6.5 for lighter-load (50% MVC) and heavier-load (80% MVC), respectively. *Stuart et al. (2018)* reported values of 6.3 for heavier-load (80% MVC) and 8.0 and 8.3 for lighter load (50% MVC) for males and females, respectively. However, perceived discomfort values from the present study using an isometric task were more similar (CAFF = 8.0 and PLA = 7.2 for 30% MVC and CAFF = 7.5 and PLA = 7.4 for 70% MVC). Previous literature has reported significantly greater values for pain and exertion for an isometric task using 50% MVC compared to a dynamic task using 75% MVC when performed to momentary failure (*Frey Law et al., 2010*). This suggests the relationship for discomfort might be different for isometric compared to dynamic muscle actions. Nonetheless, despite not finding statistically significant differences for perceived discomfort, we also did not find that RPD was statistically equivalent in the lighter-load condition whereas it was for the heavier-load condition. The standardised effect size for comparison of RPD between caffeine and placebo in the lighter-load condition ($d = 0.543$ ($-0.110$ to $1.196$)) was not too dissimilar form the standardised effect size found for TTF ($d = 0.489$ ($-0.169$ to $1.146$)). But, contrastingly to the anticipated analgesic effects of caffeine, this suggested that during the

lighter-load condition participants rated higher perceptions of discomfort as a result of caffeine supplementation. These, results question whether there may be either an effort reducing, or an analgesic effect of caffeine that results in improved endurance under conditions of higher discomfort (i.e. lighter-loads) but this should be examined in further research.

### Limitations

It would be remiss not to discuss the limitations of the present study. Primarily, we did not include a non-placebo control condition and as such we cannot be certain that a placebo effect was not evident. In addition, we did not assess the effectiveness of blinding upon participants who might have experienced typical sensation of increased arousal following caffeine consumption compared to placebo. Indeed, *Saunders et al. (2017)* suggest that supplement identification can influence exercise performance. Furthermore, we did not measure blood caffeine concentrations and thus the provision of a specific dose (100 mg) combined with the potentially different absorption rates as a result of oral contraceptive or genetic factors cannot be discounted. A further limitation is the sample size in our study; however, we have presented a larger participant number than many previous studies, and we believe the study design along with the sample size fairly represents the results obtained. Based on the contemporary thinking surrounding a gene-caffeine interaction (*Guest et al., 2018*), as well as the possibility that an oral contraceptive pill might impact the effects of caffeine upon the body (*Abernethy & Todd, 1985*), future research might consider genetic testing as well as consideration of the menstrual cycle and contraceptive use; something which we failed to do. As such a larger sample size might become redundant in females where assessment of the effects of caffeine could be measured based on individual variables. Also, we accept that larger caffeine doses (3–6 mg·kg$^{-1}$ or larger) might produce more notable positive effects upon strength (*Pickering & Kiely, 2019*). However, our aim was not to identify optimal dose or current best practice, but rather fill a gap within the literature (*Spriet, 2014*) by considering whether, and how, low caffeine dosage impacts strength and muscular endurance at heavier-and lighter-loads. In addition, as noted we did not measure RPE or RPD during the exercise bouts and thus the lack of effects at task failure may miss any effect on the exercise bout. Lastly, we did not measure other possible mechanisms through which the effects of caffeine are thought to impact maximal strength and muscular endurance. Thus, the effects upon MVC seen here may be due to unmeasured mechanisms such as blocking the binding of adenosine to $A_1$ and $A_{2a}$ G-protein coupled receptor sites and increased sarcoplasmic reticulum calcium release impacting upon the central nervous system and motor unit recruitment respectively (*Grgic et al., 2019b*).

### CONCLUSIONS

In the present study of females, we found a statistically significant effect of low dose caffeine (100 mg, equating to ~1.50 mg·kg$^{-1}$) upon knee extension MVC, though the meaningfulness of this performance enhancement is less clear. Further, we did not find statistically significant effects of low dose caffeine upon TTF, or RPE and RPD. Indeed,

RPE was statistically equivalent for isometric knee extension to momentary task failure in all conditions examined, and RPD for the heavier-load condition. As discussed herein, the dosage used represents an ecologically valid approach (e.g. a specific absolute dosage rather than a personalised relative dose per kg of body mass), and further, is representative of typical caffeinated beverages. Since caffeine consumption can present with some deleterious side effects and withdrawal symptoms, we suggest a low caffeine dose (similar to that found in typical caffeinated beverages) might be suitable practical approach to attain enhancements in muscular strength, though not relative muscular endurance.

### Funding
The authors received no funding for this work.

### Competing Interests
The authors declare that they have no competing interests.

### Author Contributions
- Georgina Waller conceived and designed the experiments, performed the experiments, analyzed the data, authored or reviewed drafts of the paper, and approved the final draft.
- Melissa Dolby conceived and designed the experiments, performed the experiments, analyzed the data, authored or reviewed drafts of the paper, and approved the final draft.
- James Steele conceived and designed the experiments, analyzed the data, prepared figures and/or tables, authored or reviewed drafts of the paper, and approved the final draft.
- James P. Fisher conceived and designed the experiments, performed the experiments, analyzed the data, prepared figures and/or tables, authored or reviewed drafts of the paper, and approved the final draft.

### Human Ethics
The following information was supplied relating to ethical approvals (i.e. approving body and any reference numbers):

Solent University Health, Exercise and Sport Science Ethics Committee granted Ethical approval (ethical application reference: wallg2018).

### Data Availability
Raw data and code are available as Supplemental Files.

### Supplemental Information
Supplemental information for this article can be found online at http://dx.doi.org/10.7717/peerj.9144#supplemental-information.

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
