# Peer review of "A low caffeine dose improves maximal strength, but not relative muscular endurance in either heavier-or lighter-loads, or perceptions of effort or discomfort at task failure in females"

_PeerJ, doi:10.7717/peerj.9144_

## Round 0.1 · original submission · Major Revisions

Expert reviewers generally report positive comments regarding the manuscript. However, several suggestions should be considered by the authors as detailed below.

Reviewer 1 ·

Basic reporting

No comment.

Experimental design

No comment.

Validity of the findings

No comment.

Additional comments

This is a very interesting study that addresses an important gap in the literature: the effects of caffeine on exercise performance among females while using low caffeine doses. I have a few minor suggestions that would like to see addressed before acceptance.

Introduction:
Line 66: one narrative review that explored the effects of caffeine in resistance exercise provided these recommendations as well:
https://link.springer.com/article/10.1007/s40279-018-0997-y
This might be of interest to the authors as the review is quite recent. Otherwise, no comments on the Introduction. Well-written overall.

Methods:
Line 141: were the participants fasted for the testing sessions? Please specify the time of day at which the testing was performed and specify if the time of day was standardized for each participant.

It seems that the effectiveness of the blinding was not explored. If not, please add this as a study limitation, as shown by Saunders et al.
https://www.ncbi.nlm.nih.gov/pubmed/27882605

Discussion:
264: the mean effect size value is indeed a larger effect than what is previously reported in the meta-analyses that explored the effects of caffeine on strength. However, I think it would be worth noting that 95% CIs in the present study overlap with the 95% CIs in the meta-analyses that explored the effects of caffeine on strength.
https://www.ncbi.nlm.nih.gov/pubmed/20019636?dopt=Abstract
https://www.ncbi.nlm.nih.gov/pubmed/29527137?dopt=Abstract
Given the overlap of 95% CI, this might indicate the effect in the population is somewhat lower than the effect observed in this analyzed sample. I believe that adding discussion about these points is worthwhile.

Line 285: please reference the meta-analysis by Polito et al. here as well. https://www.sciencedirect.com/science/article/abs/pii/S0765159716000563

Line 300: please specify that RPE was assessed after completing the test and note that there is evidence supporting lower RPE following caffeine ingestion when assessed during exercise.
https://www.ncbi.nlm.nih.gov/pubmed/15773860

Limitations section
If the time of day was not standardized, this is another limitation worth specifying.

Figures:
Very nice figures, even though they might benefit from higher resolution.

·

Basic reporting

• Clear, unambiguous, professional English language used throughout.
Yes.

• Intro & background to show context.
It is well and clearly written. I do suggest to the authors to justify why women should have different response from caffeine consumption than men.

• Literature well referenced & relevant.
Yes.

• Structure conforms to PeerJ standards, discipline norm, or improved for clarity.
Yes.

• Figures are relevant, high quality, well labelled & described.
In figures 1 to 4, change caffiene for caffeine.

• Raw data supplied
Yes.

Experimental design

• Original primary research within Scope of the journal.
Yes.

• Research question well defined, relevant & meaningful. It is stated how the research fills an identified knowledge gap.
Yes.

• Rigorous investigation performed to a high technical & ethical standard.
Yes.

• Methods described with sufficient detail & information to replicate.
Yes.

Validity of the findings

• This study has higher degree of novelty and practical information.

• This study showed a null effect of ~1.5 g/kg caffeine supplementation over RPE, RPD and TTF (regardless of load), and performance improvement for MVC. The audience should be aware about this information.

• Meaningful replication encouraged where rationale & benefit to literature is clearly stated.
Yes.

• All underlying data have been provided; they are robust, statistically sound, & controlled.
Yes.

• Speculation is welcome, but should be identified as such.
Addressed.

• Conclusions are well stated, linked to original research question & limited to supporting results.
Yes.

Additional comments

The authors did a great work with practical information.
I have minors concerns about this study, as follow:
• Title: I do suggest withdrawing “…as well as muscular endurance…” because the statistical analysis did not support this statement;

• Introduction: I do suggest including a sentence (or paragraph) justifying why women should have different outcomes than men;

• Statistical analysis: Although it was a great and detailed work, the authors could improve the information for the audience including the homogeneity, normality and sphericity analysis;

• Results: fix the figures (1-4) (change caffiene for caffeine);

• Discussion: TTF – form line 279 to 283 – I do suggest deleting this statement. I understood the authors’ thought. However, the descriptive data has too large standard deviation which disagree with the speculation/analysis done. It will impact in the study’s title and the last sentence in the conclusion;

• Conclusion: I do suggest deleting the last sentence “Since caffeine consumption can present with some deleterious side effects and withdrawal symptoms we suggest a low caffeine dose might be suitable to attain enhancements in strength and muscular endurance.” Because the data does not agree with it.

·

Basic reporting

The manuscript presented aims to assess the potential ergogenic effects of acute low dose caffeine consumption on muscular strength in females. It is evident that the work has been meticulously conducted and the written communication is generally well presented however the research rationale and key findings need to be better contextualised with respect to the current state of the literature. The manuscript offers an important insight into a number of recently identified contemporary issues in this field by firstly examining the effects of a relatively low dose (200mg) caffeine (below the typically cited 3mg.kg threshold believed to induce an effect on physical performance) and by adding to the dearth in published articles examining the effect of caffeine supplementation in females.

Experimental design

The research question is clearly communicated and the experimental design appropriate to address the aims of the study. Some further work is needed to better contextualise the need to answer the question proposed and there are some minor amendments are required to improve the clarity of the method. Please see specific comments below.

Validity of the findings

The data provided and the statistical approach is adequate with the discussion and conclusion well focused on the primary results. As with the introduction section, more work is needed to better contextualise these results and the broader application of these findings could be more explicitly presented. See specific comments below.

Additional comments

Abstract
- In the method, please indicate what muscle group was tested at what angle the tests were performed. It would also be worth further indicating the characteristics of the population, (i.e. trained or untrained; habitual caffeine users?)
- In the results section please be consistent in the number of decimal places when reporting values.
- Lines 85-86. It is not clear if this sentence relates to the effect of caffeine or indicates a difference between the two tests? The effects of caffeine on 70% MVC are unclear.
- Line 42. The sentence starting here is awkward, please reword.

Introduction
- The first two paragraphs should be shortened and amalgamated into one to provide a more precise overview of the topics specifically related to the area.
- I’m not sure that the argument around being a ‘considerable gap’ in literature with respect to exploring caffeine effects on muscular strength still holds given the number of recent systematic reviews on this subject. Would it not be better to state that despite such evidence there is a larger amount of conflicting findings for this mode of exercise and that caffeine effects are likely influenced by, participant characteristics, dose, exercise modality, muscle group?
- Line 59 – is ‘side-effects’ the correct term here?
- There needs to be a more robust and critical account of previous evidence exploring the effects of caffeine in females in order to better contextualise the need for this work. Most importantly there is a need to acknowledge the reasons for why there is limited work in females and the potential pitfalls in experimental design that may influence the results.
- The section on line 67 needs some further thought. An optimal dose is likely to be individual and exercise specific. The rationale for examining low dose caffeine (and specifically 1.5mg.kg) needs to be more explicitly presented. The reviews by Pickering et al (https://doi.org/10.1016/j.nut.2019.06.016) and Spriet et al (https://doi.org/10.1007/s40279-014-0257-8) maybe useful.
- Line 78 – this has been considered in an isolated muscle model (https://doi.org/10.1007/s12576-012-0247-20
- Line 96 – it would appear that your study has a number of differences to the study cited. Consider removing this here and incorporating into the review of the literature above in order to improve clarity

Method
- Line 113. It would be useful to cite the mean and standard deviation for the typical caffeine consumption to categorise the participant’s in to low or high habitual caffeine consumers,
- Line 126 – What angle was the MVC completed at? Why was this chosen?
- Did you measure or control for contraception/phase of the menstural cycle? If not this is a limitation of your study and the potential impact of this on the findings should be considered in further detail in the limitations section
- Line 131 – given your dosing strategy I think you need to be clear in the abstract and at relevant parts throughout that you are actually examining the effects of 200mg of caffeine and that 1.5mg.kg is the average relative dose. In its current form this is a little misleading.
- Line 148 and 149 appear to be a repeat
- Was MVC assessed during all four experimental trails? Is so it is not clear how this was analysed in the outline of the statistical method
- Line 193 – please provide a reference to support the thresholds used to denote large and moderate effect sizes

Results
- The figures are well presented and it is nice to see that you have opted to report the individual reposes. Please check that the quality of the images aligns to the requirements of the journal.

Discussion
- Line 242 replace ‘and’ with ‘an’, add to this first paragraph by providing a summary of the key findings
- The paragraph starting on line 246 seems a little out of context, a more concise version of this would be useful in the introduction to better justify the dose. Also consider are the examples provided relevant. Is it likely that athletes are going to consume a Starbucks prior to physical activity?
- The discussion of MVC data needs to be better contextualised with respect to previous work examining the effect of caffeine on strength and caffeine effects in female participants
- TTF – please outline how these results compare to other studies examining the effect of caffeine on repeated muscular contractions
- If perception of effort/pain did not account for the results presented, can the authors suggest what the likely underlying mechanism(s) may be.
- A more explicitly account of the applications of this data for practice would be useful
- Conclusion – the negative side effects of caffeine consumption should be discussed earlier in the manuscript to better contextualise the application of the data presented

---

## Round 0.2 · accepted · Accept

All reviewers are satisfied with the changes implemented in the manuscript. Congratulations!

Reviewer 1 ·

Basic reporting

No comment.

Experimental design

No comment.

Validity of the findings

No comment.

Additional comments

Great work with the revision. Congrats on this paper.

·

Basic reporting

No comment

Experimental design

No comment

Validity of the findings

No comment

Additional comments

I am pleased with the article's improvement. Nothing to add.

·

Basic reporting

Please see previous review.

Experimental design

Please see previous review.

Validity of the findings

Please see previous review.

Additional comments

The authors have adequately addressed the comments outlined in my original review.